# Higher Integrin Alpha 3 Beta1 Expression in Papillary Thyroid Cancer Is Associated with Worst Outcome

**DOI:** 10.3390/cancers13122937

**Published:** 2021-06-11

**Authors:** Lorenza Mautone, Carlo Ferravante, Anna Tortora, Roberta Tarallo, Giorgio Giurato, Alessandro Weisz, Mario Vitale

**Affiliations:** 1Department of Medicine, Surgery and Dentistry, University of Salerno, 84081 Baronissi, Italy; lorenza.mautone@gmail.com (L.M.); carlo.ferravante@unina.it (C.F.); anntortora@unisa.it (A.T.); rtarallo@unisa.it (R.T.); ggiurato@unisa.it (G.G.); 2Genomix4Life, 84081 Baronissi, Italy; 3Genome Research Center for Health-CRGS, University of Salerno Campus of Medicine, 84081 Baronissi, Italy

**Keywords:** integrins, thyroid cancer, BRAF mutation

## Abstract

**Simple Summary:**

Integrins are cell-extracellular matrix adhesion molecules considered functionally related to the development of cancer metastasis. Starting from a large dataset of mRNA-seq of papillary thyroid carcinoma (PTC), we investigated the potential role of integrins in the clinical course of PTC patients. Results showed that the PTC “classical” and “tall cell” histology variants display a more similar integrin expression profile with respect to the ‘follicular’ variant. Interestingly, the BRAFV600E mutation was found to be associated with a higher expression of integrins compared to RAS mutations. The integrin subunit ITGA3 was associated with advanced disease stage, lymph node metastasis, extrathyroidal extension, high-risk, and a worst prognosis. In vitro assays with PTC cell lines demonstrated the role of the α3β1 integrin in cell motility and invasion, evidence that supports the role of this adhesion molecule in tumor progression. These results demonstrate the existence of a PTC-specific integrin expression signature that correlates with histopathology, specific driver gene mutations, and aggressiveness of the disease.

**Abstract:**

Integrins are cell-extracellular matrix adhesion molecules whose expression level undergoes quantitative changes upon neoplastic transformation and are considered functionally related to the development of cancer metastasis. We analyzed the largest mRNA-seq dataset available to determine the expression pattern of integrin family subunits in papillary thyroid carcinomas (PTC). ITGA2, 3, 6, V, and ITGB1 integrin subunits were overexpressed in PTC compared to normal thyroid tissue. The PTC histology variants “classical” and “tall cell” displayed a similar integrin expression profile with a higher level of ITGA3, ITGAV, and ITGB1, which differed from that of the “follicular” variant. Interestingly, compared to RAS mutations, BRAFV600E mutation was associated with a significantly higher expression of integrins. Some integrin subunits were associated with advanced disease stage, lymph node metastasis, extrathyroidal extension, and high-risk groups. Among them, ITGA3 expression displayed the highest correlation with advanced disease and was associated with a negative prognosis. In vitro scratch assay and Matrigel invasion assay in two different PTC cell lines confirmed α3β1 role in cell motility and invasion, supporting its involvement during tumor progression. These results demonstrate the existence of a PTC-specific integrin expression signature correlated to histopathology, specific driver gene mutations, and aggressiveness of the disease.

## 1. Introduction

Survival of many eukaryotic cell, including epithelial cells, requires appropriate interactions between adhesion molecules and the extracellular matrix (ECM). These interactions are deeply altered in cancer cells, enabling them to invade surrounding tissues, survive, and form secondary tumors [1].

Integrins are the predominant cell surface receptors of ECM proteins such as fibronectin (FN), laminin, vitronectin (VN), and collagen (CO) [2]. These transmembrane heterodimers are composed of a common β1 chain noncovalently associated with a distinctive α-subunit, whose level of expression undergoes quantitative changes upon differentiation, neoplastic transformation, and hormone or cytokine induction [3,4]. At least 18 α and 8 β subunits are known in Humans, generating 24 heterodimers, many of which recognize FN, VN, CO, and several other large glycoproteins. Integrins bind to ECM cluster in subcellular structures known as focal adhesions. These sites provide a structural link between the actin cytoskeleton and the extracellular matrix and control signal transduction pathways that modulate proliferation, differentiation, survival, migration, and tumorigenesis in many cell types.

Thus far, integrin expression in normal and pathological thyroid cells has been analyzed mainly by immunodetection and RT-PCR, but their role in thyroid tumorigenesis has been poorly investigated. Alpha3 β1 and αVβ3 are the most abundant integrins in normal thyroid cells, and to a lesser extent α1β1, α2β1, and α6β1 [5,6,7,8,9]. These receptors' expression is modulated by cell status (i.e., cell-to cell contact) and tumor transformation [10]. Increased expression of α2, α5, α6, and β4 integrin subunits have been documented in different thyroid tumor histotypes, while α4 gave inconsistent results [5,6,11,12,13,14]. Besides structural functions bridging ECM to the cytoskeleton, integrin activation has additional crucial functions in normal and neoplastic thyroid cells. For example, αVβ3 binding to FN is necessary for survival and promotes the proliferation of normal thyrocytes [15,16,17]. Several studies in cell models and primary tumors demonstrated the biological role and potential clinical impact of integrins in thyroid tumors. Expression of integrin subunit β4 correlates with lymph node metastasis in papillary carcinomas, and integrin subunit β6 significantly correlates with recurrence of follicular thyroid carcinoma (FTC) [18,19]. Inhibition of α5β1 binding to ECM prevents attachment of FTC cell lines to the bone matrix [20], while RET-induced cell adhesion and migration of papillary thyroid carcinoma (PTC) cell lines required β1 and β3 integrins in vitro and in a mouse tumor xenograft model [21]. Finally, knock-down of β4 subunit expression reduced the proliferation, migration, and anchorage-independent growth of anaplastic thyroid carcinoma cells in vitro and xenograft tumor growth in vivo [22].

The availability of gene expression large datasets generated by RNA sequencing provided us the opportunity to extensively investigate the expression of integrin subunits in thyroid cancer samples. Results demonstrate that integrin gene expression hallmarks PTC subtypes and that higher α3β1 expression is associated with a worst outcome of the disease.

## 2. Materials and Methods

### 2.1. Samples Collection

The analysis has been performed considering dataset TCGA-THCA, downloaded from The Cancer Genome Atlas (TCGA) (https://portal.gdc.cancer.gov/ accessed date 21 April 2020). TCGA-THCA dataset comprised raw-counts data with the gene expression profile of 496 PTC tumor samples and 58 solid tissue samples: 358 classical Papillary Thyroid Carcinoma (PTCcl); 37 Tall Cell PTC (PTCtc); 101 Follicular Variant of PTC (PTCfv). Matching clinicopathological data were downloaded from cBioPortal for Cancer Genomics (http://www.cbioportal.org/ accessed date 21 April 2020) [23]. Raw-counts data with the gene expression profile of 243 PTC samples where lymphocytes and stromal cells contamination was assessed were downloaded from TCGA (https://portal.gdc.cancer.gov/projects/TCGA-THCA accessed date 21 April 2020).

### 2.2. Data Analysis

Data analysis of TCGA-THCA data has been performed as described by Salvati et al. [24]. In detail, HTseq counts data from TCGA-THCA project were imported in R package DESeq2 v1.26.0 (R version 3.6.3) [25]. Differential expression was reported as fold change ≥ |1.5| along with associated adjusted *p* values (FDR ≤ 0.05), computed according to Benjamini–Hochberg [26].

### 2.3. Cell Lines, Antibodies, Immunofluorescence and Flow Cytometric Analysis

TPC-1 and BCPAP (Leibniz Institute DSMZ, Braunschweig, Germany) were papillary thyroid cancer stable cell lines harboring the RET/PTC and BRAFV600E oncogenes, respectively. The cells were cultured at 37 °C, 5% CO_2_ in DMEN, 10% calf serum (CS).

Monoclonal antibody to α3 was purchased from Santa Cruz Biotechnology (Dallas, TE, USA); fluorescein-conjugated anti-mouse antibody was from Sigma-Aldrich (St. Louis, MI, USA). Cells were then analyzed by flow cytometry using a FACScan apparatus (Becton Dickinson Co., Mountain View, CA, USA). Flow cytometric analysis was performed as follows: cells were harvested by trypsin-PBS, incubated with the primary monoclonal antibody for 1 h at room temperature (RT) in 0.5% BSA-PBS, washed with the same buffer, and incubated again with the secondary fluorescein-conjugated antibody for 30 min at RT. Cells were resuspended in BSA-PBS and analyzed by flow cytometry.

### 2.4. Downregulation of α3 Expression by siRNA

The cells were transfected with 3 unique 27mer ITGA3 siRNA duplexes or scrambled negative control siRNA duplex (10 nM) following the manufacturer protocol (OriGene, Rockville, MD, USA). Residual α3 expression was assessed 48 h after transfection by flow cytometry with anti-α3 antibody. In BCPAP and TPC-1 cell lines, residual α3 expression was 30.04% +/− 3.1 and 29.5% +/− 3.8, respectively (mean of 4 independent experiments).

### 2.5. Cell Attachment Assay

The assay was performed in 96-well flat-bottomed microtiter plates. The wells were filled with 100 μL of the appropriate dilution in PBS of FN, CO, or LM (Collaborative Research, Bedford, MA, USA). After overnight incubation at 4 °C, the plates were washed with PBS, filled with 100 μL 1% heat-denatured BSA, and incubated for 1 h at room temperature. Then, plates were washed and filled with 100 μL/well PBS, 0.9 mmol/L CaCl 2, and 0.5 mmol/L MgCl 2 containing 70 × 10^4^ cells. After 1 h at 37 °C, plates were gently washed 3 times with PBS, and the attached cells were fixed with 3% paraformaldehyde for 10 min, followed by 2% methanol for 10 min, and finally stained with 0.5% crystal violet in 20% methanol. After 10 min, the plates were washed with tap water, the stain was eluted with a solution of 0.1 mol/L sodium citrate, pH 4.2, in 50% ethanol, and the absorbance at 540 nm was measured by a spectrophotometer. In the adhesion inhibition assay, the cells were co-incubated with 1 μg/mL anti-α3 monoclonal antibody in plates previously coated with 5 μg/mL FN, CO, or LM. Alternatively, the adhesion assay was performed with cells transfected with ITGA3 siRNA or scrambled negative control siRNA. All experiments were performed in quadruplicate.

### 2.6. In Vitro Scratch Assay

Cell plates were coated with ECM with a mix of PBS 100 μg/mL LM, FN, and CO incubated for 24 h at 4 °C. TPC-1 or BCPAP cells were seeded to obtain a confluent monolayer. After 24 h, a scratch was created with a p200 pipet tip, and the anti-α3 integrin subunit monoclonal antibody was added to a fresh medium at a 1 μg/mL concentration. Alternatively, the cells were transfected with ITGA3 siRNA or scrambled negative control siRNA, plated on ECM coated plates and, after 24 h, the scratch was created. The cells were photographed, and scratches were measured by ImageJ software (https://imagej.nih.gov accessed date 10 November 2020) after 24 h. After 48 h of transfection, residual α3 expression was assessed by flow cytometry with anti-α3 antibody.

### 2.7. Invasion Assay

2 × 10^5^ cells were plated in the upper chamber on a 24-well transwell polycarbonate membrane with 8.0 μm pore size (Corning, New York, NY, USA), previously coated with 50 μL of 3 mg/mL Matrigel (Corning) and kept at 37 °C for 24 h to ensure solidification. Cells were placed in 150 μL of serum-free DMEM, 0.1% BSA, whereas the bottom chamber media contained 10% CS. After 48 h, the cells on the top of the membrane were carefully removed by a cotton-tipped applicator. The cells on the bottom of the membrane were fixed with 3% paraformaldehyde for 10 min, followed by 2% methanol for 10 min, and finally stained with 0.5% crystal violet in 20% methanol. After 10 min, the membranes were washed with tap water, the stain was eluted with a solution of 0.1 mol/L sodium citrate, pH 4.2, in 50% ethanol, and the absorbance at 540 nm was measured by a spectrophotometer.

### 2.8. Statistical Analysis

All data were presented as mean ± standard deviation. We compared groups using the Wilcoxon test, and statistical significance was defined as a *p*-value less than 0.05. Comparisons of integrin expression and clinical features were conducted by one-way analysis of variance (ANOVA), and statistical significance was assumed for *p* < 0.01. Univariate regression analysis was performed using the Spearman rank correlation test.

## 3. Results

### 3.1. Effect of Non-Tumoral Cells on Integrin Expression Assessment

Some PTC samples of the TCGA dataset were contaminated by lymphocytes and stromal cells. Both these cells have their own integrin repertoire that can interfere with the correct assessment of integrin expression in tumoral cells. To determine the effect of lymphocytic infiltration, we compared the counts of PTC samples with <15% of lymphocytes (*n* = 130) vs. those with 15–40% (*n* = 8) (Appendix A). Only the integrin subunits known to be expressed by leukocytes (ITGA 4, D, L, M, X, ITGB2) were significantly higher in samples contaminated by lymphoreticular cells. Similarly, to determine the effect of stromal cells, we compared the counts of PTC samples with <95% of stromal cells (*n* = 60) vs. those with >30% (*n* = 53) (Appendix A). Only the integrin subunits ITGA1 and ITGA5 were significantly higher in samples contaminated by stromal cells. This analysis indicated that the counts of ITGA 2, 3, 6, V, and ITGB1 referred to the mRNA expression of PTC tumor cells and were not affected by the presence of lymphocytic or stromal contamination.

### 3.2. Expression of Integrin Subunits mRNA in Normal Thyroid Tissue

Analysis of the TCGA-TCA database confirmed the results previously obtained by immunodetection in normal thyroid tissue. ITGA3, ITGAV, and ITGB1 were the most abundant integrin subunits, followed by ITGA1 and ITGB3 (Figure 1). Very low counts of ITGA5 and ITGA6 mRNA were present. The combination of these α and β subunits corresponded to some well-documented heterodimers, detected by immunodetection (α3β1, αVβ3, α1b1 and α5β1). Notably, the expression of main subunits mRNA was quite constant among samples, as documented by their low standard deviation (α3 = 52%; αV = 37%, β1 = 28%; β3 = 53%). PTC displayed an increased expression of ITGA2, 3, 6, V, and ITGB1 (*p* < 0.001).

### 3.3. Comparative Analysis of Integrin Subunits in PTC Variants

TCGA included a large repository of RNA-seq data relative to PTC, comprising classical (PTCcl), follicular (PTCfv), and Tall Cell variants (PTCtc). The analysis of integrin mRNA expression of this dataset allowed a comparative analysis between PTC variants (Figure 2). PTCtc displayed higher expression of ITGA2, ITGA3, ITGAV, and ITGB1 subunits. PTCfv displayed a significantly lower mRNA expression of ITGA2, 3, V, and ITGB1 compared to PTCcl and PTCtc. Between PTCcl and PTCtc, only ITGA3 and ITGAV had a *p* < 0.001. Thus, the integrin expression profile highlighted a similarity among PTCcl and PTCtc that distinguishes them from PTCfv.

### 3.4. Association of Integrin Expression and Driver Gene Mutations

Of the 321 PTCcl analyzed, the BRAFV600E mutation was detected in 205 and mutated RAS in 43. In general, the expression profile of integrin subunits in BRAF mutant, RAS mutant, and non-BRAF non-RAS mutant PTC was similar, with higher expression levels in BRAF mutant tumors (Figure 3). Variance analysis of the three tumor groups demonstrated a significant difference in all subunits except for ITGA 1, 5, and 6. The largest difference occurred between BRAF mutants and RAS mutant PTCcl. The higher ratios BRAF mutants/RAS mutants among α subunits were observed for ITGA2 and ITGA3, with a fold-change of 4.2 and 3.3 (*p* < 0.0001), respectively. Cumulative analysis of all three variants of PTC produced identical results.

### 3.5. Association of Integrin Subunits Expression and Clinical Features

We analyzed the correlation between integrin subunits expression and the Thyroid Differentiation Score (TDS), a single metric parameter designed by the expression level of 16 thyroid metabolism and function genes [27]. ITGA1 positively correlated with the TDS, while ITGA2, ITGA3, and ITGA4 negatively correlated with the TDS, with ITGA3 being the most significant (R = −0.71, *p* < 0.0001) (Figure 4A, Table 1). Advanced stages were associated with integrin subunits: stage IV with ITGA 3 and stage III with ITGA 2 (Figure 4B, Table 2). The expression patterns of most integrin subunits were associated with lymph node metastasis (Table 3), extrathyroidal extension (Table 4), and high risk of recurrence (Table 5). In addition, in this case, the correlation was higher for ITGA3 (Figure 4C,D). High risk of tumor recurrence, according to the 2015 American Thyroid Association guidelines [28], was associated with higher ITGA3 expression, whereas higher expression of ITGA2 and ITGAV correlated with intermediate-risk (Figure 4E, Table 5). Disease-free survival analysis showed that tumors with higher expression of ITGA2 or ITGA3 had a shorter time between tumor recurrences. Tumors with ITGA2 above the median had a hazard ratio for disease recurrence of 1.9, *p* = 0.039. Tumors with ITGA3 expression above the median had higher risk of recurrence log-rank *p* = 0.013, hazard ratio = 2.1, *p* = 0.015 (Figure 4F).

### 3.6. Adhesion of Cancer Cell Lines to ECM

Two well-characterized papillary thyroid cancer cell lines, TPC-1 and BCPAP, harboring the RET/PTC and BRAFV600E oncogenes, respectively, were used to investigate the role of the integrin α3β1 in adhesion to ECM [29]. Cell attachment assays were performed in microtiter plates coated with different concentrations of FN, CO, or LM (Figure 5A). Both cell lines displayed a weak adhesion to LM compared to FN and CO. Integrin α3β1 mediated the adhesion to LM as demonstrated by the inhibitory effect of the anti-α3 antibody, while it was ineffective on adhesion to FN and CO (Figure 5C). The cells transfection with ITGA3 siRNA oligo duplexes 48 h before the adhesion assay reduced α3 expression to 30% (Figure 5B) and inhibited LM adhesion to 59% (Figure 5C).

### 3.7. Integrin α3β1 and Motility in Thyroid Cancer Cell Lines

In vitro scratch assay was used to investigate the role of the integrin α3β1 in cell motility. The cells were plated in ECM coated wells and, after 24 h, a scratch was created and measured immediately (Figure 6). The scratch was no more visible in untreated cells or after transfection with scrambled RNA (CTRL). The presence of an anti-α3 antibody added to the medium immediately after the scratch inhibited cells motility to 49% in BCPAP and 65% in TPC-1. The cells transfection with ITGA3 siRNA oligo duplexes 24 h before plating (48 h before the scratch was produced) reduced α3 expression to 30% and inhibited cells motility to 19% in BCPAP and 28% in TPC-1.

### 3.8. Invasion Assay

BCPAP and TPC-1 were plated on top of a polycarbonate membrane coated with Matrigel, and cells migrating to the lower face of the membrane were assessed after 48 h. The cells were incubated with or without Ab anti α3 at a final concentration of 1 μg/mL (upper and lower chamber). In separate experiments, the cells were transfected with ITGA3 siRNA or scrambled RNA (Figure 7). In both cell lines, Matrigel invasion was inhibited by the anti α3 antibody and by ITA3 siRNA.

## 4. Discussion

Reported results of gene expression analysis here confirmed previous immunodetection studies in normal thyroid cells. Considering α and β subunits assembly in the natural heterodimers, only α3β1, αVβ3, and at a minor extent α1β1 and α6β1 receptors are expressed in normal thyroid cells. Thus, integrin repertoire in normal thyroid cells appears restricted but sufficient to enable the cell to bind FN, CO, and LM, besides other non-ECM proteins. These integrins have functional and structural properties, participating in thyroid cell survival and follicle structure. The αVβ3 receptor for FN and VN is a functional component of the cell-ECM interaction and plays a central role in FN-mediated survival in thyroid cells through complex signaling, involving p21 RAS, calcium/CaMKII, and PI3K/AKT [8,16,17]. The α3β1 integrin interacts with ECM proteins, including members of the LM family [30]. Among the five distinct LM isoforms, α3β1 preferentially binds to LM-10/11, a major basement membrane component of the thyroid follicle [31]. The distribution on the cell membrane of these two integrins is different in the normal thyroid cell. Alpha Vβ3 is clustered in the focal adhesions while α3β1 is spread on the entire basal membrane in a polarized fashion, being located exclusively on the basal plasma membrane and at intercellular contact sites [7,8].

Tumor transformation deeply alters the equilibrium between adhesion molecules and basal membrane. Invasive carcinomas generally show aberrant basement architecture and loss of LM around the epithelial structures [32]. Early thyroid immunohistochemical studies highlighted that adenomas and non-invasive FTC tend to preserve the basement membrane around the follicles while its partial or complete loss can be recognized in widely invasive FTC [33]. In contrast, PTC preserves basement membrane structure with abundant LM around the follicles [31]. More extended is the change of integrin expression in thyroid tumors. The polarized distribution of α3β1 and αVβ3 integrins on the cell membrane, a feature of normal thyroid cells, is lost while other integrin subunits are expressed de novo both in benign and malignant tumors [34]. The loss of polarized integrins expression on the cell surface inadenomas and carcinomas highlights a common biological alteration concerning cell-stroma interaction.

Analysis of the TCGA database highlighted a defined difference between PTC subtypes with higher integrins expression in PTCcl and PTCtc, differing from PTCfv lower integrin expression. Thus, with respect to the integrin profile, PTCcl is closer to PTCtc than to PTCfv. However, when the TCGA database was generated, the histology subtype non-invasive follicular thyroid neoplasm with papillary-like nuclear features (NIFTP) had not yet been described. Therefore, a number of NIFTPs are present in the PTCfv and this may be responsible for the different integrin profiles of this variant compared to the others.

BRAF mutation was associated with higher expression of all integrin subunits and particularly to ITGA2, ITGA3, and ITGAV, contributing to the difference between integrin expression profiles of the three PTC histotypes, as BRAF mutation was the driver mutation in 64% and 82% of PTCcl and PTCtc and only 17% in PTCfv.

The α2β1 complex is the one with higher upregulation in thyroid carcinomas (up to 9-fold for ITGA2 vs. 2.5-fold for ITGA3 and ITGAV), although it is not the most abundant integrin expressed. This receptor, whose expression in benign and malignant thyroid tumors was previously detected by immunodetection, binds LM, COL, and heparan sulfate proteoglycan core protein, an integral component of the basement membrane [6,33]. Alpha 2β1 is over-expressed in a variety of cancer cells and experimental evidence suggest that its aberrant expression might contribute to invasion, metastasis, and drug resistance in non-small cell lung, ovarian, and breast cancer, pancreatic ductal adenocarcinoma, and hepatocellular carcinoma, being associated with poor survival in pancreatic ductal adenocarcinoma and potential therapeutic target for gastric cancer [35,36,37,38,39]. Moreover, α2β1 has been investigated as a potential therapeutic target in thyroid tumors. In TPC-1, a PTC cell line, ropivacaine suppresses proliferation, invasion, migration, and accelerate apoptosis via regulation of ITGA2 expression [40]. In our analysis in PTC, ITGA2 is positively associated with stage III, lymph node metastasis, minimal extrathyroidal extension, and intermediate-risk tumors.

ITGA3 is the most expressed integrin subunit and the most strongly associated with aggressive disease, being found upregulated in stage IV, lymph node metastasis, moderate/advanced extrathyroidal extension, high-risk tumors, and shorter disease-free survival. Recently the role of ITGA3 in thyroid carcinogenesis has been investigated. Knockdown experiments in TPC-1 demonstrated that ZNF367, a member of the zinc finger protein family, regulates cellular adhesion, invasion, and migration through, at least in part, modulation of ITGA3 expression [41]. In TPC-1 cells, miR-524-5p inhibits cell viability, migration, invasion, apoptosis, and autophagy through targeting ITGA3 [42]. These and many correlation studies point to integrin α3β1 playing a role in thyroid carcinogenesis.

Invasion and metastasis are complex processes that require sequential events: adhesion, proteolysis of ECM, and migration. In principle, well-differentiated epithelial tumors commonly use collective migration mechanisms [43]. Collective migration requires a subset of highly mobile cells at the front that generate migratory traction. Cells at the leading margin engage integrins in anterior protrusions towards the ECM, whose components are degraded by surface proteases. For example, the serine integral membrane peptidase seprase, MMP1 (a collagenase), and MMP2 (a gelatinase) bind, respectively, α3β1, α2β1, and αvβ3 integrins [44,45]. A positive correlation between integrin function and invasion was shown for many tumors, including melanoma, ovarian, colorectal, and prostate cancer. The data reported here demonstrate that differentiated thyroid carcinomas similarly display a selective increase in integrin expression and a positive correlation with the disease aggressiveness.

## 5. Conclusions

Integrin expression profile is altered in PTC and correlates with histopathology subtype and specific driver gene mutations.

Integrin α3β1 is involved in PTC cell motility and invasion and is associated with advanced-stage disease, lymph node metastasis, extrathyroidal extension, high-risk, and a poor prognosis.

## Figures and Tables

**Figure 1 cancers-13-02937-f001:**
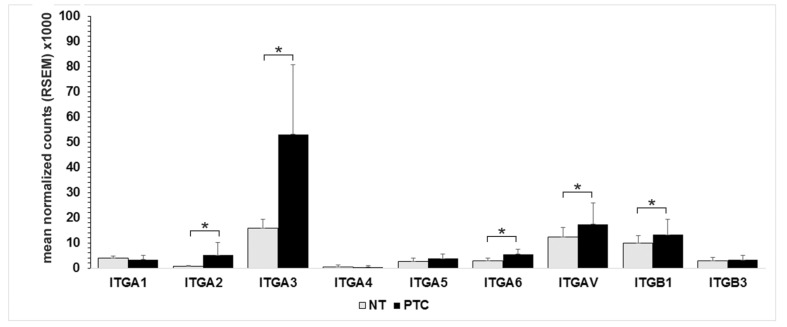
Integrin subunits expression in TCGA database. Mean and standard deviation of mRNA integrin subunits of papillary thyroid carcinoma PTC (black bars, *n* = 321) and normal thyroid tissue (NT, gray bars, *n* = 58). * Wilcoxon *p* < 0.001.

**Figure 2 cancers-13-02937-f002:**
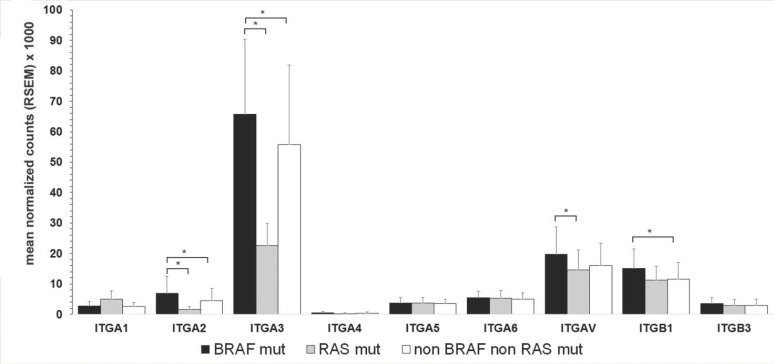
Integrin subunits expression in PTC variants. Mean and standard deviation of mRNA integrin subunits of PTC classical (PTCcl, *n* = 358), follicular (PTCfv, *n* = 101), and tall cell variants (PTCtc, *n* = 37). * Wilcoxon: * *p* < 0.001.

**Figure 3 cancers-13-02937-f003:**
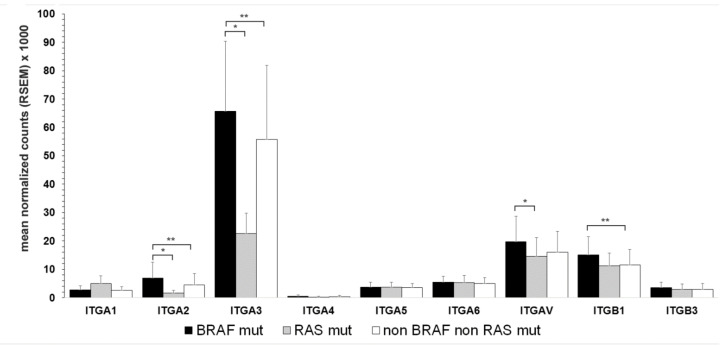
Correlation between integrin expression and driver mutations in PTCcl. Integrin subunits expression were determined in PTCcl with BRAF mutation (BRAF mut), RAS mutation (RAS mut) or none of both (non BRAF non-RAS). Mean and standard deviation; ANOVA *p* < 0.01 for all subunits except for ITGA 4, 5, and 6. Wilcoxon: * *p* < 0.0001, ** *p* < 0.001.

**Figure 4 cancers-13-02937-f004:**
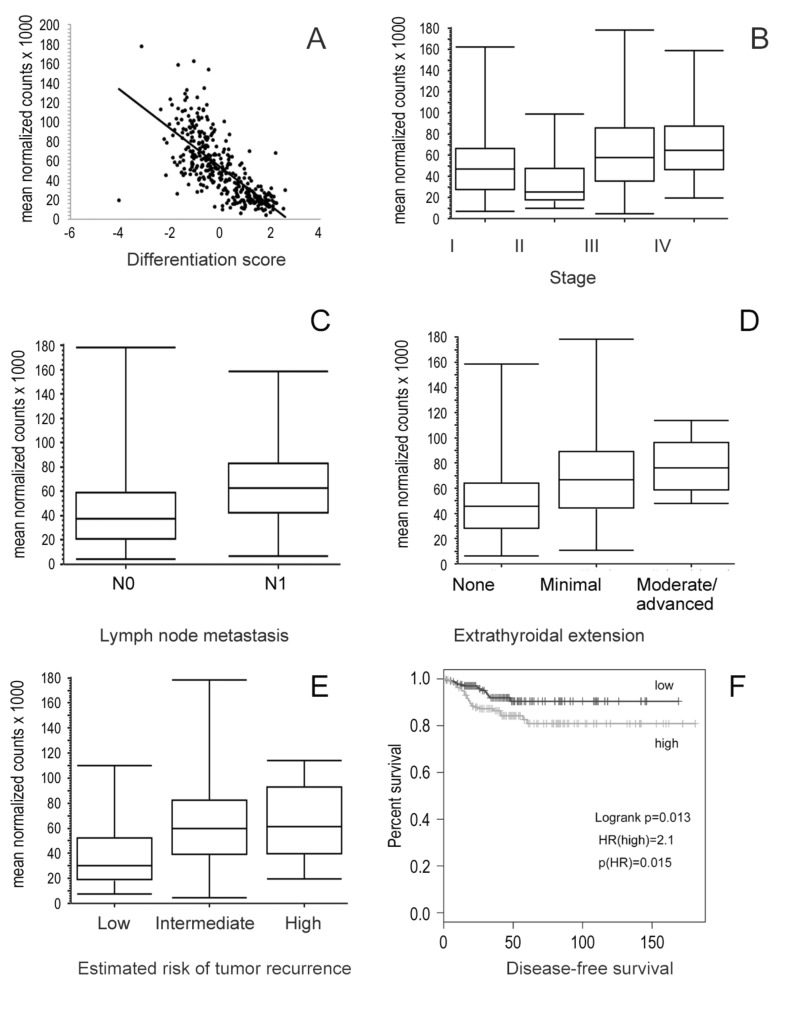
Correlation between ITGA3 and differentiation score (**A**), disease stage (**B**), lymph node metastasis (**C**), extrathyroidal extension (**D**), and estimated risk of tumor recurrence (**E**). For all analysis *p* < 0.0001. (**F**) Kaplan–Meier disease-free survival plot for ITGA3. Low and high refer to the median mRNA expression. Log-rank *p* = 0.013, hazard ratio = 2.1, *p* = 0.015.

**Figure 5 cancers-13-02937-f005:**
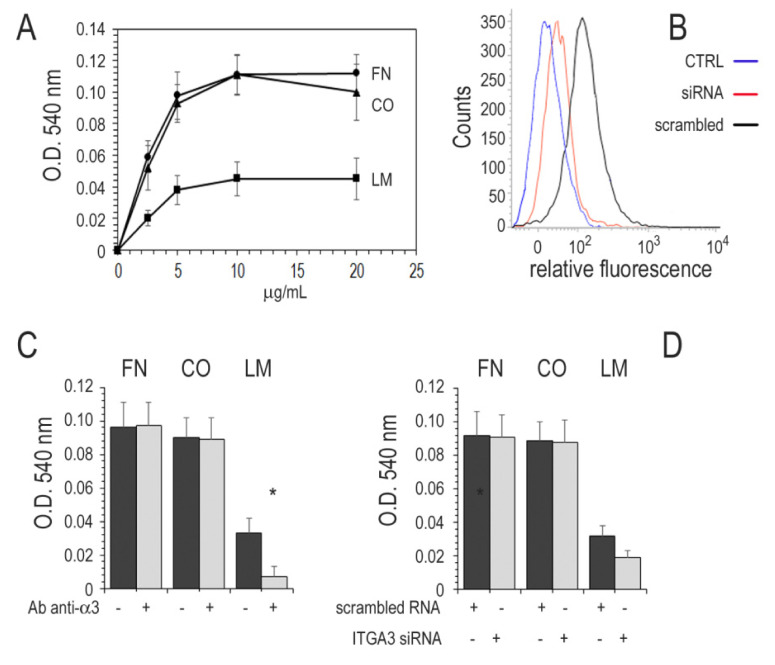
Cell attachment to ECM and role of α3β1. (**A**) Microtiter wells were coated with different concentrations of FN, CO, or LM and saturated with heat-denatured BSA. Cells in calcium-and magnesium-containing PBS were added and incubated at 37 °C for 1 h. The attached cells were measured as described in Materials and Methods. (**B**) The cells were transfected with ITGA3 siRNA duplexes (siRNA) or with scrambled RNA, and α3 expression was assessed after 48 h by flow cytometry with anti-α3 primary antibody (CTRL, no primary antibody). The mean inhibition of α3 expression of 4 experiments was about 70%. (**C**) Microtiter wells were coated with 5 μg/mL ECM and saturated with BSA. Cells were added to the plates with 1 μg/mL of monoclonal antibody anti α3 subunit (Ab anti-α3). (**D**) Alternatively, an attachment assay was performed with cells transfected with ITGA3 siRNA or scrambled RNA. After 1 h at 37 °C, the plates were gently washed, and attached cells were measured as described. All experiments were performed in quadruplicate. *, *p* < 0.001. Only experiments with BCPAP are shown, results in TPC-1 were all the same.

**Figure 6 cancers-13-02937-f006:**
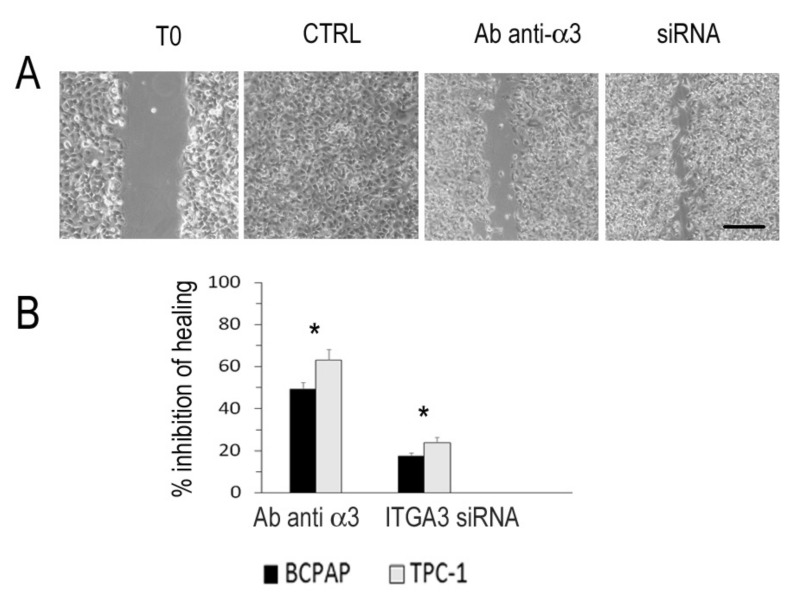
Scratch assay. BCPAP (**A**) and TPC-1 cells were plated in ECM coated wells, a scratch was produced (T0), and the healing process was photographed after 24 h. The cells were untreated or transfected with scrambled oligos (CTRL), transfected with ITGA3 siRNA duplexes, or incubated with monoclonal antibody anti-α3 subunit (Ab anti-α3). Scale bar, 200 μm. Magnification ×200. (**B**) Hedges distances were measured and reported as percentage of inhibition of healing. Standard deviations of triplicate experiments were <5%. *, *p* < 0.001.

**Figure 7 cancers-13-02937-f007:**
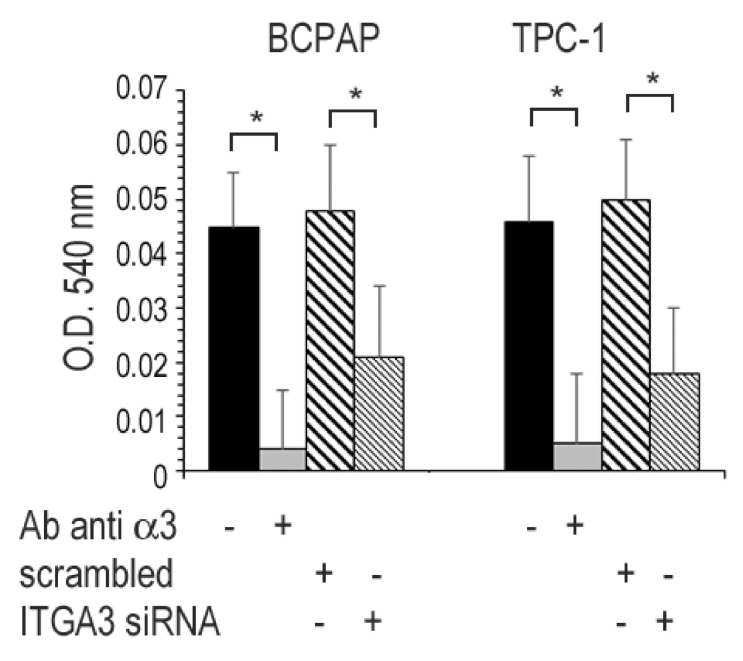
Matrigel invasion assay. Cells were plated on a polycarbonate membrane coated with Matrigel, then cultured for 48 h. The cells on the top of the membrane were carefully removed, and attached cells on the bottom of the membrane stained by crystal violet and measured as described in the Materials and Methods. The cells were incubated with or without Ab anti α3 integrin subunit or, in separate experiments, the cells were transfected with ITGA3 siRNA or scrambled RNA. All experiments were performed in quadruplicate. *, *p* < 0.0001.

**Table 1 cancers-13-02937-t001:** Spearman analysis of the correlation between integrin subunits and Differentiation score as described in [27].

Subunit	R	*p*
ITGA1	0.51	<0.0001
ITGA2	−0.50	<0.0001
ITGA3	−0.71	<0.0001
ITGA4	−0.17	0.0043
ITGAV	−0.25	0.0024
ITGB1	−0.24	0.0034

**Table 2 cancers-13-02937-t002:** Mean gene expression and ANOVA analysis of the correlation with disease stage. Only significant data are reported. Stage IV included both IVA and IVC.

	STAGE				
Subunit	I	II	III	IV	*p*
ITGA1	3437	4551	2917	2688	<0.0001
ITGA2	5179	3198	5695	5133	0.0417
ITGA3	50620	35950	60690	68806	<0.0001

**Table 3 cancers-13-02937-t003:** Mean gene expression and Wilcoxon analysis for lymph node metastasis. Only significant data are reported.

Subunit	N0	N1	*p*
ITGA1	3637	2894	0.0001
ITGA2	4051	6251	<0.0000
ITGA3	44,777	64,078	<0.0000
ITGAV	15,126	19,600	<0.0000
ITGB1	11,763	14,709	<0.0000

**Table 4 cancers-13-02937-t004:** Mean gene expression and ANOVA analysis of the correlation with extrathyroidal extension. Only significant data are reported.

	Estrathyroidal Extension			
Subunit	None	Minimal	Moderate/ Advanced	*p*
ITGA1	3632	2770	2426	<0.0001
ITGA2	4284	7124	5963	<0.0001
ITGA3	46,027	69,104	77,692	<0.0001
ITGAV	16,035	20,639	20,250	<0.0001
ITGB1	12,503	14,808	17,491	<0.0001

**Table 5 cancers-13-02937-t005:** Estimated risk of tumor recurrence based on the 2009 American Thyroid Association guidelines.

	Risk			
Subunit	Low	Intermediete	High	*p*
ITGA1	4087	2929	2651	<0.0001
ITGA2	3321	6218	5497	<0.0001
ITGA3	37,455	62,487	65,943	<0.0001
ITGA4	342	446	269	0.02108
ITGAV	14,544	19,230	17,161	<0.0001
ITGB1	11,686	14,119	14,403	0.0003

## Data Availability

The data presented in this study are available on request from the corresponding author.

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
