# Peer review of "Higher Integrin Alpha 3 Beta1 Expression in Papillary Thyroid Cancer Is Associated with Worst Outcome"

_cancers, 2021, doi:10.3390/cancers13122937_

Round 1
Reviewer 1 Report
(1) The major paradigm-shifting after the TCGA PTC project has not been well illustrated in this study. The majority of so-called FV-PTC (with RAS signature) in the TCGA cohort is now better reclassified as a follicular neoplasm (non-invasive follicular thyroid neoplasm with papillary-like nuclear features [NIFTP] or follicular adenoma) but not a PTC variant anymore. The true "invasive" FV-PTC mostly harbors a BRAF-like signature (PMID: 30645670). In the results and conclusions of this current study, I found "The PTC histology variants classical and tall cell displayed a common integrin expression profile with higher ITGA3, ITGAV and ITGB1, which differed from that of the follicular variant" and "Interestingly, BRAFV600E mutation resulted associated with a significantly higher expression of integrins than RAS mutations" are basically the same thing. The current concept no longer supports that BRAF-like and RAS-like tumors are within the same spectrum of PTC but more likely two different tumor types. Since this study is based on an old public dataset from almost 10 years ago, this should be better addressed to avoid misleading.
(2) Minor editing issues should be corrected throughout the manuscript.
Author Response
- The major paradigm-shifting after the TCGA PTC project has not been well illustrated in this study. The majority of so-called FV-PTC (with RAS signature) in the TCGA cohort is now better reclassified as a follicular neoplasm (non-invasive follicular thyroid neoplasm with papillary-like nuclear features [NIFTP] or follicular adenoma) but not a PTC variant anymore. The true "invasive" FV-PTC mostly harbors a BRAF-like signature (PMID: 30645670). In the results and conclusions of this current study, I found "The PTC histology variants classical and tall cell displayed a common integrin expression profile with higher ITGA3, ITGAV and ITGB1, which differed from that of the follicular variant" and "Interestingly, BRAFV600E mutation resulted associated with a significantly higher expression of integrins than RAS mutations" are basically the same thing. The current concept no longer supports that BRAF-like and RAS-like tumors are within the same spectrum of PTC but more likely two different tumor types. Since this study is based on an old public dataset from almost 10 years ago, this should be better addressed to avoid misleading.
ANSWER: We totally agree with the Reviewer. Indeed, an explanation of the finding that integrin profiles of classical and tall cell PTC variants differ from that of the follicular variant is that this variant includes many NIFTP and follicular adenomas. This is now addressed in the discussion. BRAFV600E and RAS positive PTCcl histotype displayed different integrin expression, confirming that these driver oncogenes are responsible of different integrin profile. The manuscript and Figure 3 now report the analysis of PTCcl alone.
- Minor editing issues should be corrected throughout the manuscript.
ANSWER: We did our best to fix it
Reviewer 2 Report
The authors have made the necessary corrections and have performed an additional experiment to support their findings. The article has been improved and can be accepted.
Author Response
The authors thank again the Reviewer for his valuable observations that helped to correct errors and improve the study.
Reviewer 3 Report
The authors improved the manuscript significantly by including more functional data testing ITGA3 in PTC; These include PTC cell attachment and invasion analysis in PTC cells compared with PTC with reduced ITGA3 expression using blocking antibody to ITGA3 or siRNA against ITGA3. There are a couple of minor points that need to be addressed.
Minor points
- line 209; Legend to Figure 3 should be changed from “BRA mutation” to “BRAF mutation”.
- Line 345; The statistics (p<0.0001) should be plotted on Figure 7 for the readers to appreciate which group is compared for statistical difference.
Author Response
- line 209; Legend to Figure 3 should be changed from “BRA mutation” to “BRAF mutation”.
ANSWER: Done
- Line 345; The statistics (p<0.0001) should be plotted on Figure 7 for the readers to appreciate which group is compared for statistical difference.
ANSWER: Done
This manuscript is a resubmission of an earlier submission. The following is a list of the peer review reports and author responses from that submission.
Round 1
Reviewer 1 Report
- The methods for the result section 3.1 "Effect of Non-Tumoral Cells on Integrin Expression Assessment" should be clarified in M&M. Did the authors evaluate all virtual slides of the TCGA dataset to determined the lymphocytes and stromal cell %? The supplementary table 1 and 2 only showed a comparison between two cases?
- Why do you use the student's T analysis for mean gene expression and lymph node metastasis? Do you expect a normal distribution?
Reviewer 2 Report
The authors have utilized the TCGA dataset to demonstrate the association of alpha3 beta1 integrin receptor with the clinicopathological features of papillary thyroid cancer. Although the TCGA dataset analysis to evaluate the associations is interesting, the authors have not performed the right set of experiments to demonstrate similar associations through preclinical studies. They have merely performed a wound scratch assay that has not been performed with the right controls (scrambled siRNA control is missing, and the graph does not have any standard deviation).
Also, the results section 3.3 does not match with the data shown in figure 2.
Overall, the article is entirely based on an analysis of the TCGA dataset. The authors should design and perform the necessary set of experiments to demonstrate the role of alpha3 beta1 experiment in papillary thyroid cancer with appropriate controls.
Reviewer 3 Report
The manuscript by Mautone et al. provides some valuable information that more advanced papillary thyroid cancer (PTC) is associated with higher expression of integrin alpha 3 when analyzed with TCGA RNA-seq data. They further demonstrate that reducing the expression of integrin alpha 3 leads to diminished PTC mobility when tested in vitro. However, one major conclusion about the integrin expression in different types of PTC subtypes is difficult to understand given the data that they provide (see below in Major points 3). In addition, one experiment based on in vitro scratch assay appear to be insufficient of data analysis as well as support for the role of integrin alpha 3 in tumor progression. I recommend major revision of this manuscript in providing more details in their analyses and additional experiment to touch on a biological role of integrin alpha 3 as the manuscript desires. Throughout the paper, there are some errors in grammar, syntax, and misspelling that needs revision.
Major points
- Supplementary Tables 1 and 2. What was the source of data to determine lymphocyte infiltration and stromal component contamination? Please detail the information and how the cutoffs for infiltration and contamination were determined.
- Figure 1. Be more specific about the units in the y-axis. Is this FPKM? RSEM? What’s the source of the normal thyroid tissue here? Indicate the number of samples -case and normal – included in the analysis.
- Figure 2. The data interpretation for Fig. 2 is misleading. The statistical analysis shows that the ITGA2, ITGA3, ITGAV, ITGB1 expression was significantly different between PTCcl and PTCtc. And the expression for those genes were statistically different between PTCfv and PTCtc. How does the authors conclude that “the integrin expression profile highlights a similarity among PTCcl and PTCtc that distinguishes them from PTCfv” (line 162)? Unless there is a mistake in the Figure legends, the interpretation should be that PTCtc is different from the other types of tumors.
Same issues in the Abstract, Conclusion, and Discussion
- Figure 5. More thorough data presentation is needed for Fig. 5B. There is no error bars and statistical analysis to appreciate the effect of ITGA3 on PTC motility.
Minor points
- line 62; “but to date their role in thyroid tumorigenesis has been purely investigated.” -> what does the “purely investigated” mean?
- line 65; change “increase” to “increased”
- line 66; change “inconstant” to “inconsistent”
- line 160-162; in addition to the problem as indicated in major points 3, there is a grammatical error in this sentence “Analyzing the differences between PTCcl and PTCtc only ITGA3 had a p <0.001, while between PTCcl or PTCtc and PTCfv the subunits with a significant differential expression were ITGA 2, 3, V and ITGB 1.”